# Concurrent Exercise Training: Long-Term Changes in Body Composition and Motives for Continued Participation in Women with Obesity

**DOI:** 10.3390/jfmk7040110

**Published:** 2022-12-07

**Authors:** Danielle D. Wadsworth, Kameron B. Suire, Ashley Peart, Shelby Foote, Chloe Jones, Mynor Rodriguez-Hernandez, James R. McDonald, David D. Pascoe

**Affiliations:** 1Exercise Adherence and Obesity Prevention Laboratory, School of Kinesiology, Auburn University, Auburn, AL 36849, USA; 2Division of Physical Activity and Weight Management, Department of Internal Medicine, University of Kansas Medical Center, Kansas City, KS 66160, USA; 3Department of Education Western Campus, University of Costa Rica, San Ramon, CA 20209, USA

**Keywords:** sprint interval training, resistance training, exercise adherence, exercise enjoyment

## Abstract

The purpose of this project was to examine the effect of a concurrent exercise program (sprint interval training and resistance exercise) on body composition in women with obesity and factors associated with continued exercise participation following the program. Twenty women (37.1 ± 7.4 y, height = 1.63 ± 0.09 m, weight = 98.22 ± 0.22 kg, BMI = 34.2 ± 2.50 kg/m^2^) participated in a 10-week exercise intervention consisting of a sprint interval treadmill protocol and resistance training three times a week totaling 30 sessions. Body composition was measured by dual-energy X-ray absorptiometry (iDXA) at pretest, 12 weeks, and six months post-intervention. Semi-structured interviews assessed participants’ perception of the program at both 12 weeks and six months. Participants significantly reduced fat mass (*p* < 0.001), gynoid fat mass (*p* < 0.010), android fat mass (*p* = 0.003), and visceral fat mass (*p* = 0.003) at 12 weeks post-test. At six months, participants maintained their reductions in fat mass (*p* = 0.015), visceral fat (*p* = 0.040) and gynoid fat mass (*p* = 0.032). There were no significant main time effects in lean mass (*p* = 0.099) or caloric intake (*p* = 0.053) at 12 weeks or six months. Themes that emerged from the semi-structured interviews at 12 weeks reflected enjoyment in the training, increases in competence and knowledge, as well as apprehension of continuing training on their own. At six months, themes that emerged reflected overcoming barriers, competence regarding high-intensity training, and a lack of competence to engage in resistance training. Sprint interval training coupled with resistance training is a feasible exercise protocol for women with obesity and results in reduced fat mass over six months. Improving women’s competence for training is imperative for continued participation.

## 1. Introduction

Obesity is defined as abnormal or excessive adipose tissue accumulation and presents a serious global health issue [1]. In the United States from 2017 to 2018 the prevalence of obesity was 42.4%, nearly an 11% increase from the years 1999–2000 (30.5%) [2]. Women show an uninterrupted increase in obesity prevalence since 1999, and projections based on prevalence in 2015–2016 estimate that 46–52% of adult women will be overweight or have obesity by 2030 [3]. These growing statistics are extremely concerning given the many comorbidities linked to obesity such as coronary heart disease or other atherosclerotic diseases, type 2 diabetes, stroke, musculoskeletal issues, some forms of cancers, sleep apnea, and even early mortality [4,5].

It is well-documented that physical activity and/or exercise can decrease the chances of obesity and obesity-related diseases [6]. Furthermore, exercise has been shown to have a greater effect on reducing visceral adipose tissue than overall body weight [7]. This is important given the linkages of excess visceral adipose tissue to metabolic syndrome and increased chances of coronary heart disease and early mortality [8,9]. Although the many benefits of exercise are known, a large portion of the population fail to meet exercise guidelines. According to the Centers for Disease Control and Prevention, in 2018, approximately 54.2% of the United States adult population met the 2008 Physical Activity Guidelines by participating in ≥150 min of moderate-intensity or ≥75 min of vigorous-intensity of aerobic physical activity per week. Only 24.1% of the population participated in a sufficient amount of aerobic and muscle-strengthening exercises combined to meet the national guidelines [10]. Physical activity and exercise rates appear to be lowest among women, who consistently participate in lower physical activity rates than men across the lifespan [11]. Furthermore, early evidence shows that physical activity may have shown further decreases across the COVID-19 pandemic [12].

In recent years, high-intensity interval training (HIIT) has been studied in terms of the physiological effects, primarily compared to moderate-intensity continuous training (MICT). Findings suggest that this modality of training produces similar training effects in a significantly shorter amount of time [13,14,15,16]. In terms of body composition changes, a systematic review and meta-analysis found that HIIT and MICT showed similar and significant reductions in whole-body fat mass, but HIIT required 40% less training time commitment in participants who are overweight or have obesity [17]. A review of HIIT protocol studies [18], found significant changes in aerobic performance and aerobic capacity following HIIT, as well as, acute and chronic improvements in insulin sensitivity. However, the author suggested there should be more studies to look at fat loss and changes in body composition as well as how these changes track over time. Sprint interval training (SIT) is a type of HIIT training that consists of intermittent bursts of all-out exercise followed by periods of rest. A recent study examined four different types of training including MICT, SIT at all out intensity, SIT at 120% of VO_2peak_, and HIIT at 90% of VO_2peak_ on a braked cycle ergometer for 12 weeks of training [19]. Findings showed visceral fat loss following SIT all-out, SIT 120, and HIIT 90 was greater in comparison with MICT. Although, these results are promising the use of a braked cycle ergometer limits use to the general population.

Adults should also participate in a minimum of two days a week of resistance training each week to meet exercise recommendations [20]. Resistance training increases strength and muscle mass [21]; induces favorable changes in body composition [22] and is related to all-cause mortality [23]. A concurrent training program consisting of SIT and resistance training may result in positive changes in body composition. Indeed, a meta-analysis showed greater fat loss occurred when the aerobic portion of the concurrent training protocol was high intensity [24]. Furthermore, muscular hypertrophy and strength gains may be amplified in higher intensity concurrent training programs [25].

Despite the documented physiological benefits of HIIT training, acceptance of this training modality is not universal for a those who are sedentary or have obesity. Hardcastle [26] and colleagues argued SIT should not be prescribed for populations who are sedentary and/or have obesity on the premise that the vigorous nature of SIT and HIIT will produce negative affective states. As a result, this may lead to decreased feelings of motivation and competence and ultimately lead to decreased exercise participation and adherence [26]. Conversely, a few studies show support for HIIT compared to MICT [27,28,29]; however, only Kong and colleagues examined enjoyment for females with obesity reporting that HIIT was more enjoyable compared to MICT over the course of five weeks of training [29]. As to the best of our knowledge, perceptions of concurrent training consisting of SIT and resistance training have not been explored in women with obesity. Therefore, the purpose of this study was to examine the effect of a SIT and resistance training exercise program on body composition of women with obesity at 12 weeks and six months. In addition, women’s perceptions regarding the intervention and intentions for further activity were explored.

## 2. Materials and Methods

### 2.1. Participants

All procedures described herein were approved by the Auburn University Institutional Review Board (protocol # 13-268 MR 1308) and conformed to the standards set by the latest revision of the Declaration of Helsinki. The clinical trial ID number is NCT04751240. A priori sample size of 18 was computed with G power to achieve adequate power (0.80), 0.05 alpha level and a small to medium effect based upon previous literature. Participants were recruited from the local community. Prior to participation, all subjects were asked to read and sign an informed consent and complete the Physical Activity Readiness Questionnaire plus (PAR-Q+) [30].

Twenty-four sedentary, women with obesity (see Table 1) participated in a 10-week concurrent exercise intervention consisting of a SIT treadmill protocol and resistance training three times a week for a total of 30 sessions at a university laboratory. Participants were excluded from the analysis if they missed more than three training sessions. Over the course of the study three participants missed more than three sessions and one participant was in a car wreck and withdrew for medical reasons resulting in a final sample size of 20 participants with an attrition rate of 16.6%. At the six-month follow-up, five women were unable to complete the six-month post-test due to surgery (*n* = 1), moved from the area (*n* = 2), or did not respond to contact by email/phone (*n* = 2). All participants were healthy, as assessed by the PAR-Q+ and did not participate in a regular schedule of exercise prior to the intervention. The flow of study participants can be found in Figure 1.

### 2.2. Study Design

Prior to the intervention, participants completed the following pretest measures: height, weight, iDXA scan, and a dietary recall. The following week, the participants were oriented to the program for three sessions. During the orientation sessions, participants were introduced to the SIT and resistance training protocols. Feedback on form and proper techniques were provided by research staff. Participants were grouped based on time preferences and were instructed by research staff throughout the protocol. Approximately 1 to 3 participants were at the laboratory at the same time and each participant received one-on-one instruction throughout the protocol. Each session, the participants completed a SIT and resistance training protocol A or B (Figure 1). The order of the sessions for weeks 1–5 was SIT followed by resistance training protocol A or resistance training protocol B followed by SIT. For weeks 6–10, the order of the sessions was SIT followed by resistance training protocol B or resistance training protocol A followed by SIT. After the 30 sessions and at six months post-test, participants completed an iDXA scan and a semi-structured interview. The protocol did not deviate from the human subjects’ board approved protocol. Figure 1 shows the flow of the study.

### 2.3. Concurrent Exercise Program

#### 2.3.1. SIT Protocol

The SIT protocol was designed to elicit physiological responses at 90–95% of heart rate max established during pretest testing. Maximum heart rate was established in pretest testing by a Bruce protocol test on a treadmill. We utilized a treadmill in this study to mimic commercially available equipment found in exercise facilities or walking. Exercise heart rates were recorded for each stage. Estimated VO_2_max values were calculated utilizing the following formula for VO_2_max estimation for women: VO_2_max (ml·kg^−1^·min^−1^) = 4.38 × T − 3.9 where T is the total time completed, expressed in minutes and fractions of a minute. The regression formula for HR/VO_2_ and the ACSM calculations [31] for determining VO_2_ were used to decide the appropriate treadmill speed with 6% incline to elicit the targeted 95% of maximum heart rate. A 6% incline was used to provide an intensity at 95% of heart rate max, while maintaining a walking pace. The range in values for 95% HR_max_ at 6% incline ranged from 136 to 172 bpm.

The SIT protocol consisted of two (first five weeks) -to- three (last five weeks) sets of three 40 s sprints with 20 s of passive recovery between each sprint and one additional minute of recovery after each set (Figure 1). All participants walked at a 6% grade. For example, at the beginning of minute three, four, and five the participant would walk for 40 s, followed by 20 s of passive rest by straddling the treadmill belt. The participant would then rest for a full minute (minute six), followed by one to two more sets of walking. Walking sets were preceded by a warming up phase of three minutes at 2.5 mph at 0% grade. At the end of the sprints, participants walked on the treadmill at 2.5 mph at 0% grade for three minutes to cool down. As the participants progressed through the weekly intervention, treadmill speed was adjusted to maintain 95% of maximal heart rate. At baseline, 95% HR_max_ ranged from 136 to 176 based on Heart rate was measured throughout the intervention with a Polar H10 heart rate chest strap monitor that displayed heart rate in real time on the treadmill screen.

#### 2.3.2. Resistance Training Protocol

To fully meet the exercise recommendations, participants participated in a resistance training program consisting of two alternating workouts. Workout A consisted of a back squat, bench press, and bent-over row. Workout B consisted of squat jumps, lunges, standing shoulder press, back extensions, and sit-ups. The resistance exercise training program, based upon undulating periodization, was set to impose fluctuating stimuli and in turn, neuromuscular overload. The training model utilized in this study included one week of orientation, three weeks of hypertrophy training, two weeks of muscular strength training, two weeks of hypertrophy training and three weeks of muscular strength training (Figure 1). The hypertrophy phase encompassed three sets of ten repetitions at 65%, 70% and 75% of each participant’s estimated for the back squat and bench press, whereas the strength phase was composed of three sets of six repetitions at 75%, 80%, and 85% of estimated 1-rep maximum (RM) for each of the lifts. Three RM values were utilized to estimate 1-RM values through the Wathen equation [32].

### 2.4. Measures

#### 2.4.1. Anthropometrics

Participant’s weight was assessed with a calibrated electronic scale (Health O Meter 500 KL, Boca Raton, FL, USA) and to the nearest 0.1 kg, while height was measured to the nearest 0.25 inches using a stadiometer.

#### 2.4.2. Body Composition

Body composition was assessed by dual-energy X-ray absorptiometry (iDXA) (GE Healthcare Lunar; Madison, WI, USA). The iDXA utilizes a fan beam X-ray and different photon energy levels with distinctive attenuation profiles to detect different body tissue, then creates high resolution images that identify the amounts of specific distribution in the body [33]. According to previous studies, the precision error for total body mass is 0.9%, total body lean mass is 0.4% to 0.5%, fat mass is 0.7% to 0.8%, and 0.6% to 0.9% for percent body fat [34,35]. In addition, iDXA has shown strong agreement with computed tomography (CT) to measure visceral fat [36]. Total and regional [android, gynoid, visceral) fat mass (FM), percent body fat (%Fat), and lean body mass were measured by iDXA. Regional body composition areas were defined using the software provided by the manufacture. Android fat mass regions of interest was assessed from the pelvis cut lower boundary to above the pelvis cut by 20% of the distance between the pelvis and neck cuts neck. Gynoid fat mass regions of fat mass were assessed from the lower boundary of the umbilicus to a line equal to twice the height of the android fat distribution. All iDXA measurements were performed by certified personnel, and all participants fasted for eight hours prior to a scan.

#### 2.4.3. Dietary Recall

To determine that the effects of the study were not due to dietary changes, participants were asked to complete a 3-day diet recall (two weekdays and one weekend day) during pretest of the intervention and one week after the intervention. Energy intake and diet components were analyzed using open-sourced software (www.nutritiondata.com accessed on 10 January 2022). Average kcals over the three days were reported for pretest, 12 weeks, and six months post-test.

#### 2.4.4. Semi-Structured Interviews

Immediately following the exercise program and at six months post-test, participants completed a brief semi-structured interview. The interviews at the end of the program focused on the overall experience with the program, enjoyment of SIT training, enjoyment of resistance training, confidence for future exercise, and intention to be active in the future. At six months, the interviews focused on current exercise participation and barriers and facilitators of exercise participation. The flexible nature of the semi-structured interviews allowed for exploration of emerging themes and were aimed to facilitate participants’ accounts of their experiences with SIT and resistance training. A single, qualitative trained female researcher conducted all participant interviews. Each interview lasted 5–15 min, was tape recorded, and then transcribed verbatim.

### 2.5. Statistical Analysis

All analyses were performed with SPSS 26.0 (IBM; Armonk, NY, USA). The effects of the program on lean body mass, fat mass, segmental fat mass (android, gynoid, and visceral), and caloric intake were analyzed using a repeated-measures analysis of variance (RMANOVA) with an alpha level of 0.05 a priori. Sphericity was not violated (Mauchly’s Test of Sphericity *p* = 0.186).

The Framework Approach provided a systematic, thematic approach to analyzing the semi-structured interviews. Specifically, this approach systematically classifies and organizes data in terms of emerging themes and patterns and is widely used in qualitative research. A period of familiarization with the dataset by the lead researcher was followed by a process of coding whereby priori themes directed by the interview guide, unexpected emergent themes, and recurring viewpoints were identified. The accuracy of the initial themes, derived from a subset of the data, was confirmed by other members of the research team, and then used to guide the indexing of the remaining transcripts. The coding process enabled the development of lower order themes to be charted and organized into salient higher order themes that manifest within the whole dataset. At the final stage of data analysis, the derived themes were subjected to the triangulation process to develop a more comprehensive understanding of the phenomenon being studied [37].

## 3. Results

A total of 20 women completed the 30 sessions within 10 weeks and all 12-week post-measures. At six months, 15 women completed an iDXA scan and a semi-structured interview. Throughout the intervention, 95% HR_max_ target heart rate was maintained for 99.6% of the sprints over the course of the intervention.

### 3.1. Body Composition

Table 1 and Figure 2 show the body composition results for the study.

#### 3.1.1. Fat Mass

There was a main effect of time (*F*_2,28_ = 8.820, *p* = 0.001) for overall fat mass. Participants significantly decreased fat mass from pretest to 12-week post-test (*p* < 0.001) and from pretest to 6-month post-test (*p* = 0.015). There were no significant differences in fat mass between the 12-week and 6-month post-test. There was a main effect of time (*F*_2,28_ = 5.368, *p* = 0.011) for android fat mass. Android fat mass significantly decreased from pretest to the 12-week post-test (*p* = 0.003); however, there were no significant differences from pretest to the 6-month post-test (*p* = 0.056) or from the 12-week post-test to the 6-month post-test (*p* = 0.488). There was also a main effect of time for gynoid fat (*F*_2,28_ = 6.13, *p* = 0.006). Participants significantly decreased gynoid fat mass from pretest to the 12-week post-test (*p* < 0.010) and from pretest to the 6-month post-test (*p* = 0.032). Again, there were no differences between the 12-week and 6-month post-test (*p* = 0.754).

#### 3.1.2. Visceral Fat Mass

For visceral fat, three women’s BMI’s exceeded 40 mL/kg^2^ at the pretest, which is the maximum for visceral fat analysis with the iDXA and could not be included in the analysis. For the remainder of the sample, there was a main effect of time for visceral fat (*F*_2,22_ = 4.60, *p* = 0.020). Visceral fat significantly decreased from pretest to the 12-week post-test (*p* = 0.003) and from pretest to the 6-month post-test (*p* = 0.040). There was no significant difference from the 12-week to 6-month post-test (*p* = 0.871).

#### 3.1.3. Lean Body Mass

Lean mass showed no significant main effect for time (*F*_2,28_ = 2.56; *p* = 0.099). Although changes were not significant, participants increased lean mass from pretest to 12-week post-test and decreased lean mass at 6 months post-test. In addition, self-reported caloric intake showed no main effect for time (*F*_2,28_ = 3.263, *p* = 0.053). Although changes were not significant, participants’ caloric consumption decreased over the 6-month timespan.

### 3.2. Semi-Structured Interviews

#### 3.2.1. 12 Weeks

Themes that emerged from the semi-structured interviews at 12 weeks reflected enjoyment in training, increased competence, and knowledge, as well as apprehension of continuing training on their own. All 20 participants reported enjoying the SIT and resistance programs. For example, “*It was actually fun, I didn’t think I would like the treadmill, but I did*”; “*I didn’t hate working out like I normally do, I looked forward to coming and I really liked the treadmill part*”; “*I have never lifted weights before, it was so much fun, and I felt really strong*”. Most of the women expressed that enjoyment was a result of increases in competence and knowledge. For example, “*For me, after a few weeks I knew I could do it* [referring to the SIT protocol], *knowing I could complete a whole workout was such a confidence boost. Once I knew I could do it, working out became fun*”; “*I did not think I would be able to complete this program, but after the first week, I was like I can do this*”; “*After the first few workouts I saw myself improve and I was like wow, I can really do this!*” Participants also expressed an increase in knowledge, particularly about resistance training. For example, “*I had never even seen weights like this* [referring to free weights]. *I was so nervous! Now I really understand what to do, and how to do it.*” Others expressed the rest cycle of SIT training was new to them. For example, “*I didn’t know I could rest during exercise. I just thought I had to run until I fell out*”; “*The rest part was great. I can do almost anything for 40 s*”; “*I had never done any type of exercise where I could rest so this was big for me. It made it doable.*” Most participants expressed apprehension about being able to exercise on their own. For example, “*I am not sure I will be able to do this on my own. I am going to try*”; “*Not sure how this type of training will go on my own. I have a plan; just hope I can stick to it.*”

#### 3.2.2. 6 Months

At six months, 12 out of 15 participants stated that they exercised regularly. Emergent themes at six months reflected overcoming barriers, competence regarding SIT training, and a lack of competence to engage in resistance training. All 15 participants at six months reported some type of barrier including the following: weather, family and social commitments, work commitments, and a lack of motivation. Most participants identified a self-regulation plan that included: goal setting, social support, or time management to overcome barriers to regular exercise. Competence was again a theme at six months. In reference to SIT training, participants discussed their ability to adapt to different modes of SIT training. For example, “*I really took what we did on the treadmill and applied it to other things. If I was on the elliptical, I would go as hard as I could for one minute, then light for 30 s*”; “*I started walking outside because I like that better. I would use my music and walk fast on the chorus and slow on the verse*”; “*At first, I started just doing what we did here* [referring to the 10 weeks of training], *but then I started adding more sprints or walking fast for longer.*” Participants reported these adaptations because of increased competence level. For example, “*I knew I could do short bursts and I knew it was effective*”; “*I did it for 10 weeks, so why not 20 weeks?*” In terms of resistance training, most participants were not confident to maintain free weight training. Some participants stated, “*Well for the weights I would just go to the girl section of the gym, you know the little hand weights*”; “*I just didn’t feel comfortable in a gym doing weights.*” Participants were further probed as to why they did not feel comfortable with resistance training. Reponses included, “*I was comfortable here, but in the gym, I felt that everyone was looking at me. I am still overweight you know*”; “*You guys wanted me to do well. I did not get that from the gym people*”; “*The equipment was different. I just didn’t feel comfortable.*”

## 4. Discussion

The purpose of this study was to determine the effect of SIT and resistance training program on body composition outcomes in women with obesity. Our results show that SIT and resistance training was effective in decreasing overall fat mass, visceral fat and gynoid fat at 12 weeks and this was maintained at six months post-test. Android fat was significantly reduced at 12 weeks and did not return to pre-testing measures at 6 months, although the difference was not statistically significant. There were no significant changes in lean mass.

A review that examined body composition changes in HIIT training showed that 10 weeks of HIIT training can reduce body fat by ~2 kg [17]. Although we examined SIT, the results from our study are in line with this finding in that on average participants lost 2.36 kg over the 10 weeks. The review further states exercise is more effective for visceral fat loss than diet and can result in ~6% drop in visceral fat. Our results show an approximate 20% decrease at 12 weeks and at six months for visceral fat. In comparison, Zhang [19] examined the effects of four different types of training on visceral fat in women with obesity and found a 13% decrease for a SIT with all-out effort, 7.6% for SIT at 120% of VO_2peak_ and 11.8% for HIIT at 90% of VO_2peak_. Our higher percentage of change at 12 weeks may be due to the combined effect of resistance and SIT training, although the evidence for reductions in visceral fat [38] and fat mass vary [39]. Comparably, a study examining post-menopausal women utilizing a resistance and HIIT training protocol showed a 4% reduction in visceral fat over 10 weeks and 30 exercise sessions [40]. While total time for exercise sessions was similar (45–55 min), resistance and HIIT were performed in a rotating fashion and not on the same day as in this current study. Based upon our findings, SIT and resistance training conducted concurrently is effective in reducing overall fat mass and segmental fat mass in women with obesity up to six months. Furthermore, women were able to maintain lean mass throughout the intervention, which is important in body composition changes in women with obesity [17]. Further research should examine differing doses of SIT and resistance training to determine the optimal dose response.

There is debate in the literature for the acceptance of HIIT/SIT training for sedentary adults. In an opinion article, Hardcastle et al. discussed the merits of utilizing HIIT training for sedentary adults. The article suggested that high-intensity training may elicit perceived incompetence, lower self-esteem, and potential failure, which may contribute to avoidance of future exercise [26]. Our results from the semi-structured interview contradicted this perspective. For the participants in this study, engaging in a SIT protocol increased their perceptions of competence and was enjoyable. Participants reported feelings of success that they had not seen in previous attempts at exercise. For example, “*The walking fast and rest cycles were doable. I just kept saying it’s just 40 s.*” According to Self-determination Theory, feelings of competence are a form of intrinsic motivation which may lead to future participation in exercise [41]. Our results from the six-month interviews support this theory in that feelings of competence led to participation in variations of SIT, while feelings of incompetence about resistance training reduced participation in resistance training. It is important to note that our participants did engage in a structured program for 10 weeks and results may be different if participants initiated a high-intensity program on their own. However, even with 10 weeks of supervised training, participants still lacked feelings of competence in completing resistance training and factors to enhance competence regarding resistance training in women should be further explored. Enjoyment of the protocol was also a theme which may be associated with the novelty of the protocol and may enhance exercise adherence long term [42] and intrinsic motivation [41].

Hardcastle et al. [26] also suggested that self-regulation is required for HIIT training outside of the laboratory. Our results concur with this statement in that most of the participants discussed behavioral self-regulation strategies (i.e., goal setting, social support, self-monitoring) as methods to overcome barriers at 6 months. In particular, time management in the form of making exercise a priority was discussed by multiple participants. For example, “*When we were coming to the lab, I had it on my schedule, so I just kept it on my schedule and that worked for me*”; “*I scheduled exercise at the beginning of my week, and I treated it just like any other appointment I had.*” These findings are congruent with previous literature that identified self-regulation as a key mediator of change for exercise behavior [43] and associated it with higher levels of exercise adherence in women [44].

There are several limitations that should be acknowledged. The CoreScan software for the iDXA is limited to measuring visceral fat for individuals between 18.5 kg/m^2^ and 40 kg/m^2^. Our study had three participants who were not included in the visceral fat analysis due to their BMI exceeding 40 kg/m^2^ at pretest. In addition, although our sample size was adequate at the onset of the study, our 6-month analysis fell short of the a priori sample size of 18. While this was a limitation, we were able to gather data over a period of six months, which is longer than previous studies. Our study did not utilize a randomized control design. Future studies should examine SIT and resistance training in larger and diverse samples utilizing a randomized design. Lastly, results from the semi-structured interviews may only reflect opinions from our limited sample and may not be generalizable to the entire population of women.

## 5. Conclusions

Our study concluded that a 10-week concurrent exercise program consisting of SIT treadmill protocol and resistance training could successfully reduce fat mass in sedentary women. In addition, the results indicated that SIT programs can be enjoyable and can increase both competence and knowledge in women that can potentially transcend beyond the intervention. The participants of this study revealed that they were not as comfortable or secure in completing unsupervised resistance training post-intervention and future studies should focus on increasing competence related to resistance training in women.

## Figures and Tables

**Figure 1 jfmk-07-00110-f001:**
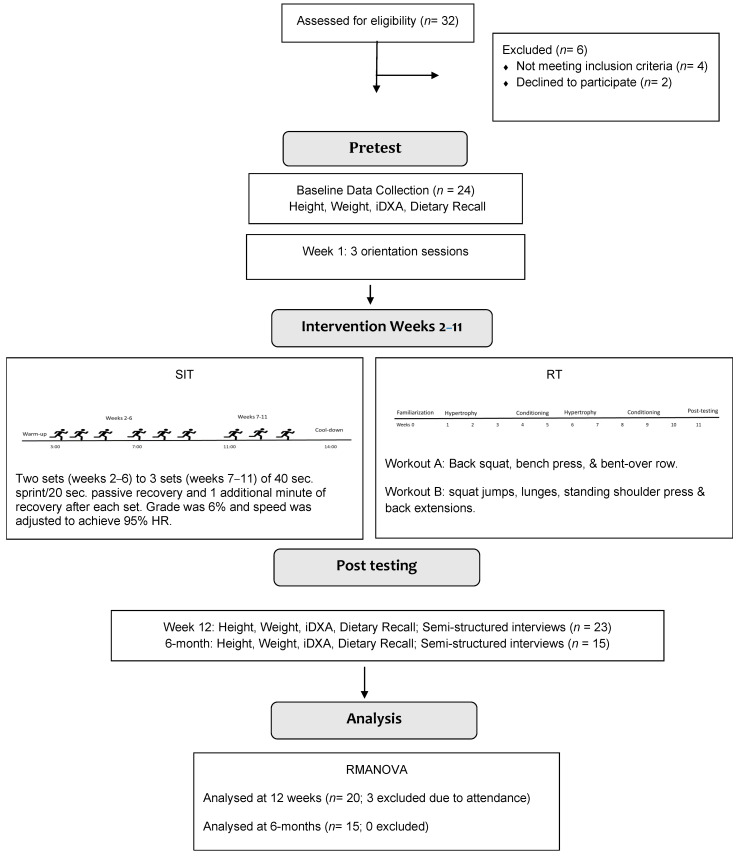
Study Flow Diagram.

**Figure 2 jfmk-07-00110-f002:**
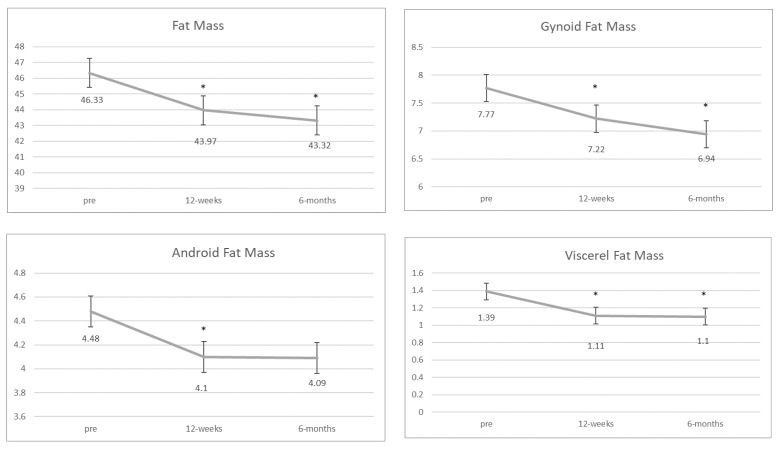
Body composition (kg) and caloric intake at pretest, 12 weeks and 6 months. * *p* < 0.05 difference is from pretest.

**Table 1 jfmk-07-00110-t001:** Demographics and Body Composition Results.

	Pre (*n* = 24) *M* ± SD (95% CI)	12 Weeks (*n* = 20) *M* ± SD (95% CI)	% Change from Pre	6 Months (*n* = 15) *M* ± SD (95% CI)	% Change from Pre			
Age (yrs)	37.10 ± 7.40							
Height (m)	1.63 ± 0.09							
Weight (kg)	97.91 ± 21.89	96.74 ± 22.55	1.19	94.54 ± 22.65	3.44			
BMI (kg/m^2^)	36.90 ± 2.50	36.40 ± 2.98	1.35	35.40 ± 3.02	4.07			
						F	*p*	η^2^
Total Fat (kg)	46.33 ± 16.78 (37.28, 55.87)	43.97 ± 16.28 (34.12, 52.16)	−5.09	43.32 ± 17.59 (33.58, 53.06)	−6.50	8.82	0.001	0.390
Gynoid Fat (kg)	7.77 ± 2.53 (6.40, 2.91)	7.22 ± 2.00 (5.88, 2.67)	−7.07	6.94 ± 2.04 (5.79, 2.63)	−10.68	6.13	0.006	0.305
Android Fat (kg)	4.48 ± 2.07 (3.36, 5.65)	4.10 ± 1.76 (3.00, 4.95)	−8.48	4.09 ± 2.04 (2.95, 5.21)	−8.70	5.37	0.011	0.277
Visceral Fat (kg)	1.39 ± 0.84 (0.85, 1.88)	1.11 ± 0.72 (0.70, 1.55)	−20.14	1.10 ± 0.95 (0.52, 1.6)	−20.86	4.13	0.029	0.256
Lean Mass (kg)	48.42 ± 7.75 (44.12, 52.72)	49.51 ± 8.81 (44.63, 54.39)	2.25	49.02 ± 8.88 (44.10, 53.95)	1.23	2.52	0.099	0.152
Caloric Intake (kcals)	2224.00 ± 432.23 (1996.78, 2389.22)	2093.00 ± 390.80 (1906.65, 2244.81)	−5.84	2085.00 ± 391.74 (1900.47, 2239.15)	−6.25	3.26	0.053	0.189

## Data Availability

All data are freely available within this manuscript.

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
