# Peer review of "Concurrent Exercise Training: Long-Term Changes in Body Composition and Motives for Continued Participation in Women with Obesity"

_jfmk, 2022, doi:10.3390/jfmk7040110_

Round 1
Reviewer 1 Report
Line 13-15 - I am assuming that there will be explanation later in the paper, but sprint training for obese women? Have they been habituated to this form of exercise?
Line 28 - this is excellent! I would encourage you to take a look at this paper and discuss it at the end of your paper as it is highly relevant to your work 10.1007/s40520-021-01853-8
Line 88 - in-text citation not provided for the Hardcastle et al.
Line 142 - the protocol seems pretty intense. I think it would be clever to have physician's clearance or medical supervision during the training sessions. I know that sample is quite young overall, but if they are not accustomed to the exercise, especially of this intensity, it might be risky.
Line 261-263 - this is an interesting finding and deserves to do be explored in the discussion.
Line 288-308 - this is great! I would take a look at this particular paper when crafting discussion. 10.3389/fpsyg.2020.577522
Line 329-331 - important and deserve to be discussed later
Line 358 - Again in-text citation for Hardcastle et al. is missing.
Overall, nice study that will be of use to both professionals and the overall population.
Reviewer 2 Report
The present study was a one-arm intervention study that examined the effect of a concurrent exercise program comprising sprint interval training and resistance exercise on body composition measured by DXA among women with obesity. In addition, the present study implemented semi-structured interviews after the 12-week intervention and at the six-month follow-up. The current issue is impressive, and the findings are unique. However, the reviewer has some concerns, as follows.
1. The authors should use People First Language in conversations about weight management.
Kyle TK, Puhl RM. Putting people first in obesity. Obesity (Silver Spring). 2014 May;22(5):1211. doi: 10.1002/oby.20727. Epub 2014 Mar 8. PMID: 24616446
2. In the introduction, the authors can update a reference about resistance training and mortality.
Momma H, Kawakami R, Honda T, Sawada SS. Muscle-strengthening activities are associated with lower risk and mortality in major non-communicable diseases: a systematic review and meta-analysis of cohort studies. Br J Sports Med. 2022 Jul;56(13):755-763. doi: 10.1136/bjsports-2021-105061. Epub 2022 Feb 28. PMID: 35228201
3. In methods, details in the setting and flow of the exercise program are unclear. For example, where and when did the participants exercise? How many participants gathered at a group session? How long was the group session? How about the order of sprint interval training and resistance exercise?
4. The explanation of SIT and RT in Figure 1 is hard to understand. Revisions are necessary.
5. What are android and gynoid fat mass? Please add the explanation.
6. “iDXA” and “iDexa” are mixed.
7. During SIT, how was the actual HR?
8. Table 1 and Figure 2 show the same results. The authors should delete either.
9. The conclusions are too long. Citing is not necessary.
Round 2
Reviewer 2 Report
I appreciate the authors for revising what I asked. The followings are the remaining concerns.
1. As previously pointed out, please show the actual HR during SIT. As reviewer 1 pointed out, the SIT protocol seems pretty intense. The authors replied to the comment that the SIT was walking at 6% grade and the HR was 95% HRmax. How much was the HRmax? If the HRmax was 180, was their HR during walking 170?
2. The reviewer thinks that Table 1 and Figure 2 must have different aspects of results. If the authors want to demonstrate a visual picture of the time change in Figure 2, Table 1 must show another element, for example, 12-week and 6-month changes and the 95% confidence interval.
3. There are some misdescriptions; hence, proofreading is necessary. For example, “95% HR” is “95% HRmax”, and “iDxa” is “iDXA” in Figure 1; “0.4” is “0.4%” in line 194; and “[” is “(” in line 196.
